# Large enhancement of response times of a protein conformational switch by computational design

Alex J. DeGrave[1], Jeung-Hoi Ha[2], Stewart N. Loh[2] & Lillian T. Chong[1]

The design of protein conformational switches—or proteins that change conformations in response to a signal such as ligand binding—has great potential for developing novel biosensors, diagnostic tools, and therapeutic agents. Among the defining properties of such switches, the response time has been the most challenging to optimize. Here we apply a computational design strategy in synergistic combination with biophysical experiments to rationally improve the response time of an engineered protein-based $Ca^{2+}$-sensor in which the switching process occurs via mutually exclusive folding of two alternate frames. Notably, our strategy identifies mutations that increase switching rates by as much as 32-fold, achieving response times on the order of fast physiological $Ca^{2+}$ fluctuations. Our computational design strategy is general and may aid in optimizing the kinetics of other protein conformational switches.

[1] Department of Chemistry, University of Pittsburgh, Pittsburgh, PA 15260, USA. [2] Department of Biochemistry and Molecular Biology, State University of New York Upstate Medical University, Syracuse, NY 13210, USA. These authors contributed equally: Alex J. DeGrave and Jeung-Hoi Ha. Correspondence and requests for materials should be addressed to S.N.L. (email: lohs@upstate.edu) or to L.T.C. (email: ltchong@pitt.edu)

The design of protein conformational switches—or proteins that adopt either 'active' or 'inactive' conformations in response to signals such as ligand binding or changes in pH—is of interest for applications that involve ligand detection and/or functional regulation. The defining properties of switches include sensitivity (e.g., ligand binding affinity), signal-to-noise ratio (e.g., change in optical or enzymatic activity of the free vs. bound states), and response time (e.g., how quickly the protein switches from one conformation to the other). For many switches, it is relatively straightforward to improve sensitivity and signal-to-noise by rational design[1–7]. These strategies are successful in large part because they can be guided by available high-resolution structures of the folded proteins that comprise each state of the switch.

Response time determines how quickly a biosensor can detect the analyte and whether it can track changes in analyte concentration in real time, and the temporal precision by which functional switches can control cellular pathways. For these applications, one typically wants the turn-on and turn-off rates to be as fast as possible. Usages that emphasize maximum signal change or sustained response instead strive for a slow turn-off rate in order to accumulate signal. It is therefore often desirable to tune the kinetics of protein conformational changes—which occur naturally over a wide range of timescales—to optimize a conformational switch for a given application. For example, $Ca^{2+}$ concentrations can fluctuate as fast as 10 ms in cells, and the slow response time of existing protein-based calcium sensors continues to hamper studies of rapid $Ca^{2+}$ signaling processes in vivo[8,9]. It is especially important to be able to accelerate the kinetics of the class of switches described here, because switching rates in this case are limited by protein unfolding events[10], and these can be slow.

Response time has proven to be the most challenging of the switch properties to improve[3,10–12], e.g., the screening of hundreds of mutant switches in one study has achieved only a fourfold improvement in the response time[3]. The reason for this difficulty is that the rational manipulation of the kinetics requires detailed views of the conformational switching pathways, including transition states that experiments typically cannot capture. Moreover, the use of atomically detailed simulations to generate pathways has not been feasible due to the long timescales of switching processes (>ms).

Here we have rationally improved the response time of a previously developed, protein-based $Ca^{2+}$ sensor[13], using a general computational design strategy in synergistic combination with biophysical experiments. Our strategy involves the use of molecular simulations that employ: (i) residue-level protein models that reproduce the expected mutually exclusive folding of individual switch components[14], and (ii) the weighted ensemble (WE) strategy[15,16], which enhances the sampling of rare events (e.g., protein folding) without biasing the dynamics. These features enable qualitative predictions of promising mutations within 2 weeks. Although transient states of the switching process of the protein-based $Ca^{2+}$ sensor under study have been generated by others[17], this was achieved by applying external biasing forces (i.e., targeted molecular dynamics and umbrella sampling), which precludes the calculation of rate constants. Furthermore, states obtained using biased and unbiased strategies can differ significantly[18]. Because the WE strategy generates statistically unbiased pathways, the resulting rate constants are rigorous, and this property allows us to predict mutations that may improve the response time of the switch.

The protein-based $Ca^{2+}$-sensor that we examine, calbindin-AFF, was engineered using the alternate frame folding (AFF) scheme[13]. This sensor is an ideal system for demonstrating the power of our design strategy due its modest size (113 residues)

and longer-than-desired response time (hundreds of ms[13]). In addition, high-resolution structures of one of the switch components, wild-type (WT) calbindin $D_{9k}$, are available[19,20]. The AFF scheme involves fusing together the WT protein and a circular permutant (CP) of the protein such that the two proteins share a portion of their sequences. This sequence overlap generates two alternate frames, N (WT structure) and N′ (CP structure) that fold in a mutually exclusive manner (Fig. 1a)[10]. In the absence of $Ca^{2+}$, these folds are populated approximately equally. A $Ca^{2+}$-binding mutation with minimal effect on the stability of WT and CP calbindin[13] was then introduced into either the N-frame (E65Q) or N′-frame (E65′Q) such that $Ca^{2+}$-binding drives conformational switching in the N → N′ or N′ → N directions, respectively. From this point on, the E65Q and E65′Q constructs will be referred to as WT E65Q and WT E65′Q constructs, respectively, since these constructs are the 'WT' constructs to which additional mutations are introduced to improve the response times of the switch. We define the forward and reverse directions of the switching processes according to the WT E65Q construct; i.e., those of N → N′ and N′ → N conformational changes, respectively.

As a proof of principle study, the application of our computational design strategy to the calbindin-AFF switch yields predictions of mutation sites that improve the response time by more than an order of magnitude. Importantly, a negative control mutation is correctly predicted to have little effect on the kinetics despite its buried location at the interface of two EF-hands where mutations might be expected to improve the response time. Furthermore, we demonstrate that the WE strategy enables efficient sampling of switching processes on a wide range of timescales such that our design strategy can be applied to switches with even slower response times. Finally, our strategy may be additionally useful to delineate transition pathways of analogous conformational rearrangements that take place in natural proteins, as part of their biological functions. Examples include photoswitchable domains[21–24] and metamorphic proteins[25,26].

## Results

**Overview of the rational design strategy.** Our computational design strategy involves a close interplay of simulations and experiments (Fig. 1b). The only prerequisite is that structures of the switch components are available, e.g., from X-ray crystallography or homology modeling. In the case of the calbindin-AFF switch, structures are available for WT calbindin (both X-ray and NMR structures)[19,20], but not for CP calbindin. However, circular permutation generally preserves the overall structure of a protein except for minor changes around the sites of permutation and linker addition, and previous NMR[27] and circular dichroism[13] results suggested that the structures of WT and CP calbindin are similar. As a further test, we compared their $^{15}N$-heteronuclear single-quantum correlation NMR spectra (Supplementary Fig. 1a). The majority of cross peaks align with the exception of amino acids 40–45 and 70–75. Mapping these residues onto the structure of WT calbindin confirms that structural differences are limited to the permutation site and the C-terminus of the protein to which the linker is attached (Supplementary Fig. 1b). We therefore constructed a model of CP calbindin in silico based on the WT calbindin structure, building in the six-residue loop that links the original N-termini and C-termini (Supplementary Methods).

Once models of the switching components have been built, the next step in the design strategy is to parameterize the residue-level simulations of WT calbindin to reproduce its experimentally measured stability. These parameters are then applied to simulations of CP calbindin, and again compared to experimental

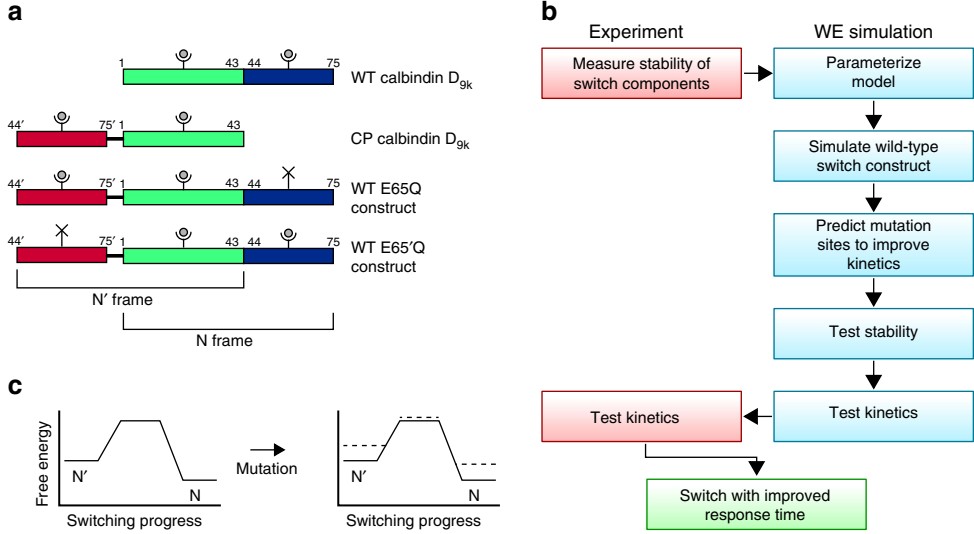

**Fig. 1** Rational design strategy for improving the kinetics of protein conformational switching. **a** Alternate frame folding scheme for the calbindin-AFF construct. Green and blue bars represent the N-terminal and C-terminal EF-hands of calbindin $D_{9k}$, respectively, and 'Y' shapes represent calcium binding sites. Duplication of the blue region (residues 44–75) and appending this segment to the N-terminus of the protein via a six-residue linker creates two alternate frames for folding that correspond to WT and CP calbindin. Following a mutation that disrupts $Ca^{2+}$-binding in one of the frames (indicated by a black 'X'), the addition of $Ca^{2+}$ preferentially stabilizes the fold of the alternate frame. **b, c** The strategy for improving the response time of the switch involves **b** a close interplay of experiment and computation to **c** obtain preferential stabilization of non-native transient states relative to the ground states (N and N′) (dashed lines), reducing the free-energy barrier for the switching process. Panel **c** depicts a schematic free-energy landscape for the E65′Q construct in the presence of $Ca^{2+}$; the strategy is applied identically to the E65Q construct, which differs in that N′ is more stable than N in the presence of $Ca^{2+}$

values, as an internal control (Supplementary Methods, Supplementary Fig. 2, and Supplementary Table 1). We then generate switching pathways via molecular simulation and use these data to predict which residues, when mutated, are likely to accelerate switching rates in both directions by preferentially destabilizing ground states (e.g., N and N′) relative to transient states, including transition states and metastable intermediates (Fig. 1c). The stabilities and kinetics of candidate mutants are tested by simulation. Tests of stability verify that the mutations destabilize, but do not unfold the corresponding switch component, while tests of kinetics provide rapid initial screening for improved response times. Finally, switching rates of the mutants are measured using stopped-flow fluorescence experiments.

**Simulation of alternate frame folding in WT switches**. Our simulations of the switching processes of the WT E65Q and E65′Q AFF constructs reproduce the following experimental observations: (i) the two alternate frames fold in a mutually exclusive manner with the $Ca^{2+}$-bound frame folding the majority of the time (as revealed by NMR experiments;[27] Fig. 2a–c), (ii) the positions at which the fluorophores were inserted in experiments are more distant in the N state than the N′ state (as revealed by stopped-flow experiments)[13] (Fig. 2d, e) and (iii) the orphaned EF-hands retain partial native-like structure in both N and N′ states (as revealed by CD experiments[6] as well as NMR chemical shift analysis;[27] Fig. 2d, e). Consistent with reports by others[28], these findings demonstrate the ability of the simulation model to capture the cooperative nature of folding processes (Supplementary Note 1; Supplementary Fig. 3). While it would be more efficient to generate switching pathways at the melting temperature of one of the switch components (i.e., 85 °C for WT calbindin)[29] rather than 20 °C, our simulations reveal that the mechanisms of switching at these two temperatures are different: at 85 °C, it proceeds through a globally unfolded state while at 20 °C,

it progresses through a partially unfolded intermediate (Supplementary Fig. 4).

To distinguish between the ground-state and transient-state conformations, we calculated free-energy surfaces based on WE simulations of the switching processes. At first glance, the free-energy surfaces for both the WT E65Q and E65′Q constructs revealed no significantly populated intermediate, suggesting a concerted mechanism (Fig. 2a–c). However, since the detection of intermediates might be obscured by unproductive pathways, we proceeded to analyze the transition path ensemble (TPE), which consists of only productive pathways each of which begins where the trajectory last exited the initial folded state and ends where the trajectory first enters the target folded state. The TPE for the WT E65Q construct again finds no intermediate, but the TPE for WT E65′Q detects the accumulation of an intermediate in a relatively shallow free-energy well, indicating a stepwise mechanism (Fig. 3a, b).

**Computational predictions to improve kinetics**. To simultaneously improve the response times of the switch in both directions, we selected candidate residues for which mutations are predicted to (i) preferentially destabilize the ground states (N and N′) over the relevant TPE, (ii) maintain sufficient stability of N and N′ (relative to the globally unfolded state, i.e., $\Delta G^{fold} \sim -2$ kcal mol$^{-1}$ in the absence of $Ca^{2+}$) such that neither state is unfolded, and (iii) retain $Ca^{2+}$ binding. We focused on large hydrophobic residues since it is more straightforward to destabilize the ground states by removing hydrophobic interactions via 'underpacking' mutations rather than stabilize the transient states by introducing non-native interactions via 'overpacking' mutations.

To address the first criterion, we quantified the contribution of each residue to the stability of the initial folded state (N or N′) relative to that of the corresponding TPE (N → N′ TPE or N′ → N TPE) (Fig. 4a; see also Supplementary Note 2 and Supplementary

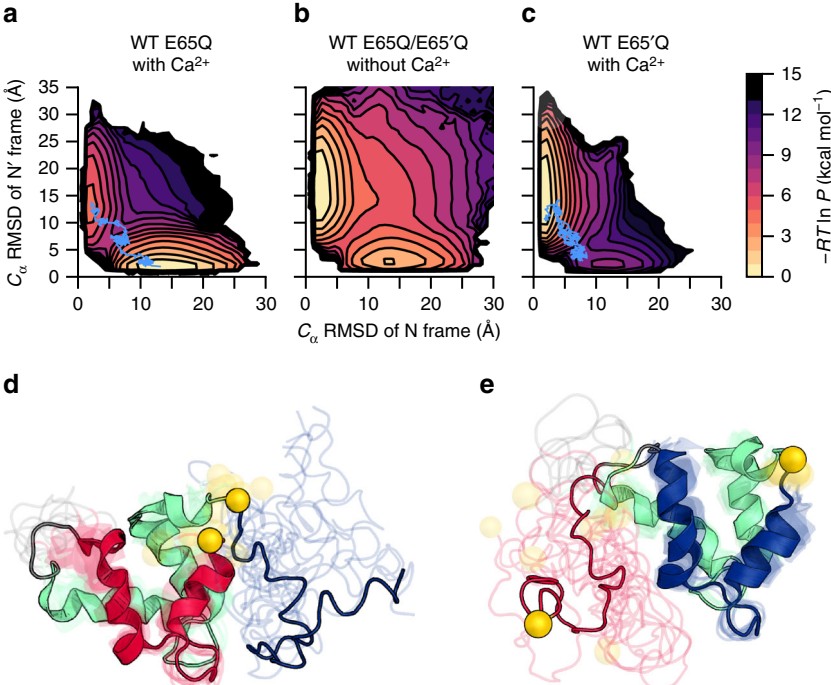

**Fig. 2** Thermodynamic and structural properties of the calbindin-AFF switch. **a–c** Free energy surfaces of the switching process as a function of the $C_\alpha$ RMSDs of the N and N′ frames from their corresponding folded conformations for (**a**) the WT E65Q construct with $Ca^{2+}$, (**b**) the WT E65Q/E65′Q construct without $Ca^{2+}$, and (**c**) the WT E65′Q construct with $Ca^{2+}$. Each free-energy surface represents a total of 20 WE simulations and thousands of independent switching pathways. The color scale represents $-RTlnP$, where $P$ is the equilibrium probability density (see Supplementary Methods). Upper left and lower right basins correspond to the N and N′ folded states, respectively. Representative transition paths in **a** and **c** are highlighted in blue and correspond to Supplementary Movies 1 and 2, respectively. See Supplementary Fig. 5 for a representative binning scheme used during the simulation of (**a-c**). **d**, **e** Representative $Ca^{2+}$-bound conformations of the N′ folded state of the WT E65Q construct (**d**) and N folded state of the WT E65′Q construct (**e**) are well-folded with partially disordered orphan EF-hands (see Fig. 1a for color scheme; yellow spheres indicate positions at which fluorophores were inserted in the stopped-flow fluorescence experiments; fluorophores were not included in the simulations)

Fig. 6 for the advantages of analyzing the TPE vs. transition state ensemble). In other words, we focused on solely the numerator of the commonly used Φ-value[30] in which the effect of a mutation on kinetics is normalized by the effect of the mutation on the relative stabilities of the initial and final states. Only the numerator is relevant to our goal of maximizing the enhancement in switching rates; furthermore, the difference in the effects of mutations on the folded states of N and N′ is very small such that Φ-values would not be meaningful. Since our simulation model is native-centric, the contribution of a residue to the stability of the state of interest (N, N′, or TPE) is the average number of native contacts formed by that residue ('contact score'; see Supplementary Methods).

As shown in Fig. 4a, the residues that exhibit the greatest differences in contact scores between the initial ground state and relevant TPE lie at the packing interface of the EF-1 and EF-2/EF-2′ hands. This location is not surprising given that the N → N′ conformational change involves undocking/unfolding of EF-2 from EF-1 and subsequent folding/docking of EF-2′ onto EF-1. Likewise, the N′ → N switch entails these same steps in reverse. Our simulations reveal additional detail in the switching mechanism, as only a subset of the residues at the interface are predicted to improve the response time when mutated. Among the residues at the EF-1/EF-2′ interface, the most promising candidates are the following four Phe residues in EF-2 (or EF-2′): F50′, F50, F63, and F66 (Fig. 4b). F50′ and F66 exhibited the greatest differences in contact scores between N′ and the N′ → N TPE, and N and the N → N′ TPE, respectively.

All four Phe residues meet the second criterion for selecting candidate mutations. Mutating each to Ala does not hamper the

ability of the N or N′ frame to fold, i.e., both simulation and experiment revealed that the mutations destabilize the analogs while maintaining a favorable folding free-energy in the absence of $Ca^{2+}$ (Table 1). In our simulations, mutations of the Phe residues were modeled as the idealized, maximum underpacking scenario in which all favorable interactions of the mutated residue with other residues were removed. All mutants unfolded reversibly and in a two-state manner in the equilibrium denaturation experiments, enabling direct comparison of experimental and computed folding free energies (Table 1 and Supplementary Fig. 7).

Finally, all four Phe residues meet the third criterion for selection, i.e., mutation would not likely disrupt binding to the $Ca^{2+}$. These side chains lie in the hydrophobic core and do not contact the side chains that coordinate $Ca^{2+}$ (E27, D54, N56, D58, and E65)[19]. While residues V61 and E60 in the EF-2 hand also exhibited large differences in contact scores between the ground-state N and the N → N′ TPE, these residues are in the vicinity of the $Ca^{2+}$-binding residues and were therefore not mutated. Mutations of F66 to Trp in the N and N′ folds (F66W and F66′W) have been tested by experiment in our earlier study and were found to decrease and increase the rate constant for the N′ → N switching process by moderate amounts (1.4-fold and 2.4-fold, respectively)[10], but this previous study did not examine the effects of underpacking mutations of F66.

An important negative control is to select a mutation that is predicted to destabilize both the ground and transition states to approximately equal extents, and to verify that the switching rates do not change. Among the residues buried at the interface of the EF-1 and EF-2/EF-2′ hands, including the promising four Phe

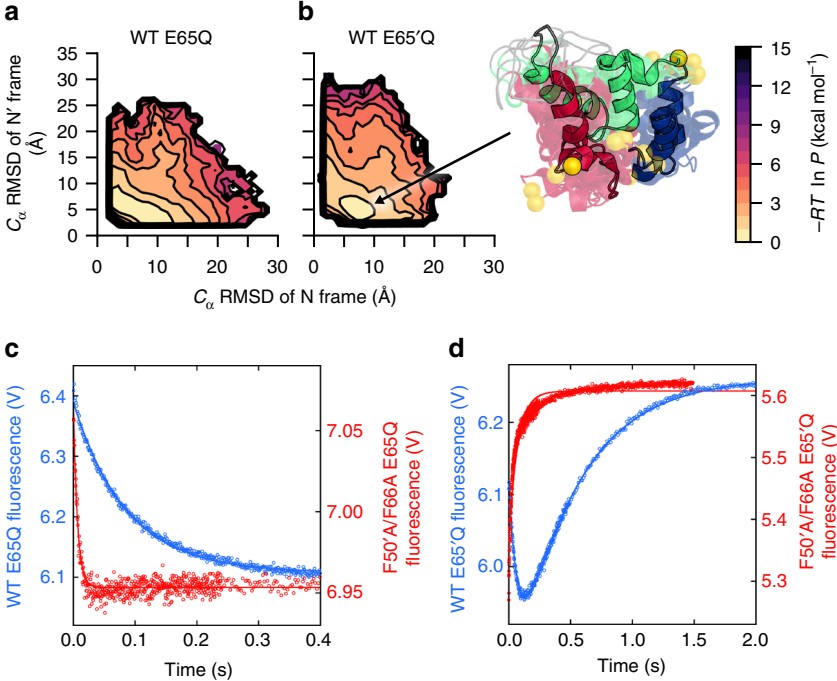

**Fig. 3** Comparison of simulated transition paths with kinetic traces from stopped-flow experiments. **a**, **b** Probability distributions corresponding to free-energy surfaces presented in Fig. 2a and c, respectively, based on only the transition path ensemble (TPE). While no intermediate exists in the probability distribution of the WT E65Q construct (**a**), a transient intermediate is evident in the probability distribution of the WT E65′Q construct (**b**, 10 representative conformations of the intermediate are shown as ribbon diagrams). The color scale represents $-RT\ln P$, where $P$ is the steady-state probability density. **c**, **d** Kinetic traces from stopped-flow experiments, in which switching is monitored by BODIPY fluorescence, reveal (**c**) single-exponential kinetics for the WT E65Q construct and (**d**) biphasic kinetics for the WT E65′Q construct. Kinetic traces are also shown for a representative set of mutations (F50′A/F66A), indicating substantial improvement to the response time of the switch. Solid lines represent a fit to a single exponential, except in the case of the WT E65′Q construct for which the data is fit to a double exponential. The rate constant for the slow, rate-limiting phase is used when assessing the fold-change in the rate constant due to mutation

residues, L31 in EF-1 exhibited the lowest change in contact scores for both the N → N′ and N′ → N switching processes and was therefore selected as the residue to mutate as a negative control. Given that EF-1 is shared between the N and N′ folds, only a single mutation needs to be introduced into calbindin-AFF, and its effect on switching rate is expected to be symmetric in each direction. As expected, the L31A mutation destabilizes the N-frame and N′-frame analogs by an extent comparable to that of the promising mutations, e.g., F50A (Table 1).

**Tests of computational predictions**. We tested the effects of the following pairs of Phe → Ala mutations on the response time of the calbindin-AFF switch: F50′A/F50A, F50′A/F63A, and F50′A/F66A. Prior to carrying out experiments of these mutant constructs, we screened the constructs by directly modeling the effects of the mutations in silico. In particular, we approximated the effects of the three pairs of Phe to Ala mutations and the Leu to Ala mutation (F50′A/F50A, F50′A/F63A, F50′A/F66A, L31A) by removing all attractive, native interactions involving the parent residue. Simulations of the switching process for the resulting mutant revealed significant increases in the rate constants for both the N → N′ and N′ → N switching processes (Supplementary Fig. 8, Supplementary Fig. 9, and Supplementary Table 2) for every promising mutant construct under consideration, whereas significant increases were not detected for the negative control L31A constructs.

Given the positive results of our computational screen, we next labeled each mutant construct with BODIPY-FL fluorophores and carried out stopped-flow fluorescence experiments to measure rate constants for the switching processes. Switching kinetics of the WT protein are similar to those measured in our earlier study[10], with one notable difference being that we now observe biphasic kinetics for the WT E65′Q construct (Fig. 3c, d). A minor phase of decreasing fluorescence, undetected in our previous study, precedes the major increasing phase that is associated with the N′ → N conformational change. It is likely that the improved optics and temporal resolution of the stopped-flow instrument used in the present study allowed us to observe this minor phase. The rate constants $k_{N'\to N}$ (representing the slower phase) and $k_{N\to N'}$ agree with our previous values within factors of 1.3 and 1.7, respectively. We attribute biphasic kinetics of the WT E65′Q construct to the accumulation of an intermediate prior to the rate-limiting N′ → N conversion. Consistent with this interpretation, our simulation detected a transient intermediate in the N′ → N direction but not the N → N′ direction (Fig. 3a, b).

Notably, all pairs of mutations increased the rate constants for both forward and reverse switching processes relative to WT—in several cases, by >10-fold (Fig. 4c, d). The largest enhancement of 32-fold was achieved by the F50′A/F50A mutant E65′Q construct. Consistent with the trend in relative contact scores of F50, F63, and F66 (Fig. 4a), the F50′A/F50A, F50′A/F63A, and F50′A/F66A mutations increased $k_{N\to N'}$ of the E65Q construct by 3-fold, 6-fold, and 12-fold, respectively, reducing the free-energy barrier by 0.7, 1.1, and 1.5 kcal mol$^{-1}$. The effects of these mutations on the N′ → N switching process were even more pronounced with 32-fold, 7-fold, and 21-fold increases in $k_{N'\to N}$ of the E65′Q construct, reducing the free-energy barrier by 2.0, 1.1, and 1.8 kcal mol$^{-1}$.

In contrast to the above results, the L31A negative control mutation increased the forward and reverse switching rates of

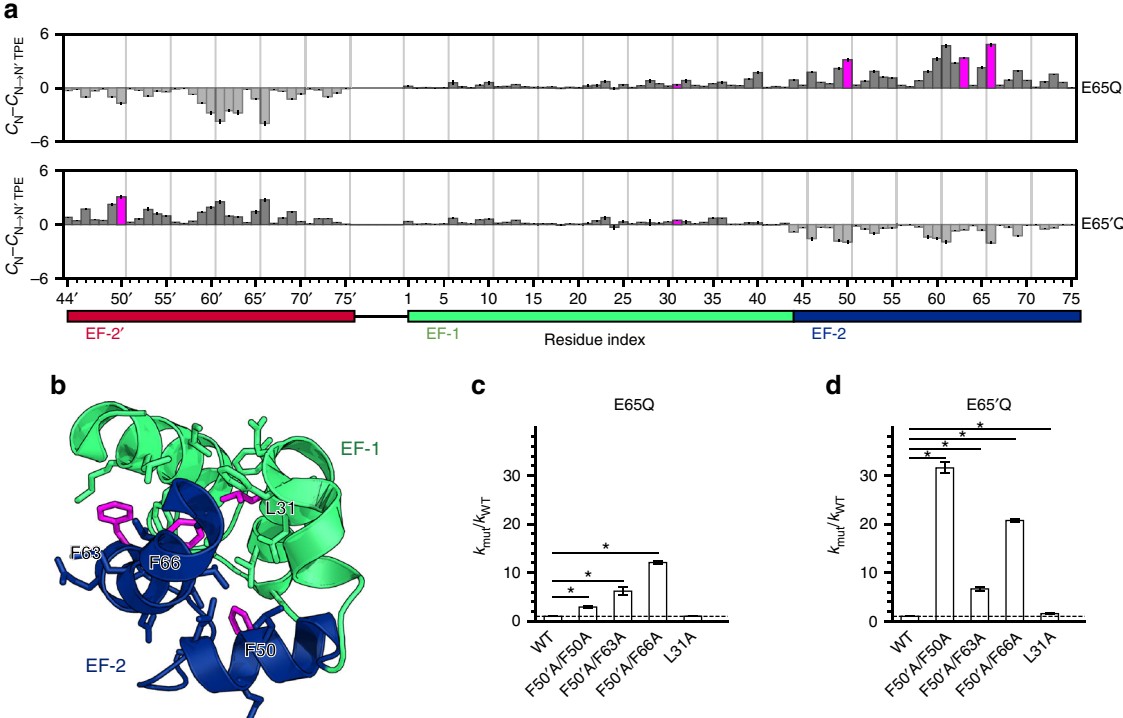

**Fig. 4** Identification and testing of mutations for improving the switch response time. **a** Residues with contact scores that are substantially lower in the transition path ensemble (TPE) than the relevant ground state (magenta bars) were identified as candidates for improving the response time (see Supplementary Methods). Error bars indicate mean ± s.e.m.; $n = 10$ independent WE simulations. **b** Residues chosen for mutation (magenta) are located at the interface of the EF-1 and EF-2 hands (green and blue, respectively), as shown in this cartoon representation of the crystal structure of WT calbindin. The displayed side chains are of residues that are in close proximity (<5.5 Å) of the residue chosen for mutation. **c**, **d** Stopped-flow fluorescence experiments reveal that all three pairs of promising mutations, F50′A/F50A, F50′A/F63A, and F50′A/F66A, increase the rate constant for switching for both the WT E65Q construct (**c**) and WT E65′Q construct (**d**), thereby improving the switch response time. As predicted for the negative control, the L31A mutation has no effect on the rate constant for the E65Q construct ($k_{mut}/k_{WT} = 1.03$; P = 0.24) and only marginally increases the rate constant for the E65′Q construct ($k_{mut}/k_{WT} = 1.57$; P = $10^{-7}$). Error bars indicate mean ± s.e.m., calculated using propagation of error; $n = 15$ stopped-flow experiments for all WT E65Q variants and E65′Q/F50′A/F50A; $n = 17$ for WT E65′Q and E65′Q/F50′A/F66A, $n = 18$ for E65′Q/F50′A/F63A, $n = 12$ for E65Q/L31A, and $n = 11$ for E65′Q/L31A. *P $< 10^{-7}$ in a one-tailed Student's t-test

**Table 1 Effects of mutations on the stabilities of N and N′ frame analogs determined by both simulation and experiment in the absence of Ca²⁺**

| | mutation | $\Delta\Delta G^{fold}_{mut-WT}$ (kcal mol⁻¹)[b] | $\Delta\Delta G^{fold}_{mut-WT}$ (kcal mol⁻¹)[b] | $\Delta G^{fold}$ (kcal mol⁻¹)[b] | $\Delta G^{fold}$ (kcal mol⁻¹)[b] | $m$ (kcal mol⁻¹ M⁻¹)[b] | $C_m$ (M)[b] |
|---|---|---|---|---|---|---|---|
| | | simulation | experiment | simulation | experiment | experiment | experiment |
| N-frame analog[a] | F50A | 1.8 ± 0.1 | 1.98 ± 0.14 | −2.1 ± 0.04 | −3.58 ± 0.04 | 1.90 ± 0.02 | 1.88 ± 0.003 |
| | F63A | 1.4 ± 0.2 | 2.86 ± 0.18 | −2.5 ± 0.1 | −2.70 ± 0.12 | 1.70 ± 0.06 | 1.59 ± 0.02 |
| | F66A | 2.5 ± 0.1 | 3.65 ± 0.16 | −1.3 ± 0.1 | −1.91 ± 0.08 | 1.43 ± 0.05 | 1.33 ± 0.01 |
| | L31A | 1.1 ± 0.2 | 1.62 ± 0.15 | −2.8 ± 0.2 | −3.94 ± 0.07 | 1.95 ± 0.03 | 2.02 ± 0.007 |
| N′-frame analog[a] | F50′A | 2.3 ± 0.3 | 1.80 ± 0.20 | −1.1 ± 0.2 | −4.08 ± 0.09 | 1.96 ± 0.04 | 2.09 ± 0.008 |
| | L31A | 1.3 ± 0.2 | 1.63 ± 0.18 | −2.1 ± 0.2 | −4.25 ± 0.04 | 2.19 ± 0.02 | 1.94 ± 0.005 |

[a] N-frame and N′-frame analogs represent WT and CP calbindin, respectively, with E65Q mutations. In experiment, the N-frame analog includes an inserted cysteine residue at position 44. The stabilities of the N-frame analog ($\Delta G^{fold} = -5.56 ± 0.13$ kcal mol⁻¹, $m = 1.88 ± 0.05$ kcal mol⁻¹ M⁻¹, $C_m = 2.95 ± 0.01$ M) and N′-frame analog ($\Delta G^{fold} = -5.88 ± 0.18$ kcal mol⁻¹, $m = 2.13 ± 0.05$ kcal mol⁻¹ M⁻¹, $C_m = 2.76 ± 0.01$ M) are similar to WT and CP calbindin (Supplementary Table 1). [b] Values indicate mean ± s.e.m. (calculated by propagation of error for $\Delta\Delta G^{fold}_{mut-WT}$); $n = 3$ independent WE simulations of folding and three independent WE simulations of unfolding for values from simulation; $n = 5$ equilibrium denaturation experiments for mutant analogs; $n = 7$ equilibrium denaturation experiments (including the two reversibility controls shown in Supplementary Fig. 7) for the WT N-frame and N′-frame analogs

WT calbindin-AFF by only 1.03-fold and 1.57-fold, respectively (Fig. 4c, d). Like the Phe residues, L31 is buried in a tightly-packed environment such that its mutation to Ala significantly destabilizes calbindin. That the L31A mutation has little effect on switching rates highlights the power of the simulations to identify rate-accelerating mutations that would not be obvious from inspecting ground-state structures.

## Discussion

While the main goal of this study was to assess the ability of contact scores derived from simulations of the WT switch constructs to predict positions where mutation improves the response time of the switch, we also assessed how well our computational screen modeled the effect of mutations. The fold-increase in $k_{N \to N'}$ calculated from simulations of E65Q variants is

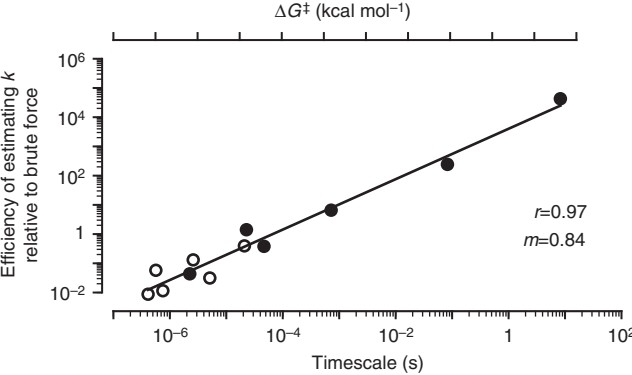

**Fig. 5** Efficiency of the computational strategy for simulating long-timescale processes. The efficiencies of WE simulations relative to brute-force simulations in computing rate constants for switching processes scale linearly with the timescale of the process, i.e., exponentially with the height of the corresponding free-energy barrier $\Delta G^{\ddagger}$ (see Supplementary Methods). Solid circles (WT constructs) represent the mean of $n = 10$ independent WE simulations, and open circles (mutant constructs) represent the mean of $n = 3$ independent WE simulations

similar to experiment (Supplementary Fig. 9 and Fig. 4c). In contrast, while simulations of the E65′Q variants correctly predict that the mutations improve the response time of the switch, the simulations do not quantitatively predict the fold-change in $k_{N'\rightarrow N}$ (Supplementary Fig. 9 and Fig. 4d). This discrepancy may be due to the fact that the $N' \rightarrow N$ switching process involves a stepwise mechanism including the formation of an intermediate while the $N \rightarrow N'$ switching process is concerted. To more accurately model the intermediate, a more detailed simulation model with the inclusion of attractive non-native interactions may be required. Nonetheless, the successful prediction via contact scores of sites where mutation improves the response time demonstrates that the simulation model used in this study is useful for protein engineering purposes, even in a case where non-native contacts might play a substantial role.

An essential feature of our computational strategy is the use of WE simulations. Relative to standard, 'brute force' simulations, the efficiency of WE simulations in calculating rate constants has been found to increase with the relevant free-energy barrier[31–36]. Here, we demonstrate that the increase is exponential with the free-energy barrier (Fig. 5; see Supplementary Methods), achieving the highest efficiencies for the uphill, reverse switching processes of the WT E65Q and WT E65′Q switch constructs (N′ → N and N → N′, respectively) that require tens of seconds and are therefore inaccessible to standard simulations. Furthermore, the WE strategy generates a greater diversity of pathways than standard simulations, yielding more statistically robust contact scores for identifying mutation sites that could improve switch response times.

In conclusion, we have applied a computational design strategy in synergistic combination with biophysical experiments to rationally enhance the response time of an engineered protein-based $Ca^{2+}$ sensor, the calbindin-AFF switch. An essential part of our approach is the use of the WE strategy in conjunction with residue-level models to greatly extend the computational reach of molecular simulations. This approach makes it feasible to generate, within days to a few weeks using a typical computer cluster, protein conformational switching pathways as slow as tens of seconds, thereby facilitating a close interplay of simulation and experiment. Based on the simulated pathways, we identified previously untested mutations that are promising for enhancing the response time of the switch via preferential destabilization of

the ground states relative to the transient states. In addition, we identified a negative control mutation that was predicted to have no effect on the response time via similar destabilization of the ground states relative to the transient states. As predicted, the negative control had little effect on the response times of the switch in both the forward (N → N′) and reverse (N′ → N) directions while all of the promising mutations substantially improved the response times of the switch in both directions. The largest improvement amounted to 32-fold, reducing the response time of the switch in the reverse direction from a mean first passage time of 590 to 19 ms, which is in range of the most rapid physiological $Ca^{2+}$ fluctuations[9]. The most effective pair of mutations, F50′A/ F66A, simultaneously increased the rate constants for both the N → N′ and N′ → N switching processes by more than an order of magnitude.

Our computational design strategy is a general one that can be applied to any protein conformational switch of a similar size (e.g., less than a few hundred amino acids) provided that the switching process occurs on the timescale of <100 s and structures of the switch components are available from either experiment or homology modeling. Furthermore, since all protein conformational switches function based on the relative stabilities of alternate conformations, our strategy is applicable to all such switches, including ones that function by other mechanisms that do not involve as large conformational transitions as the mutually exclusive folding of protein domains, provided that the expected relative stabilities are reproduced. The strategy could therefore be valuable for a variety of applications, including the design of more rapid biosensors and temporally accurate control mechanisms in synthetic biology.

## Methods

**WE simulations**. We applied the WE path sampling strategy[15] in conjunction with Brownian dynamics simulations and a residue-level protein model to efficiently parameterize the model, generate complete pathways of protein conformational switching, and calculate rate constants. In this strategy, multiple trajectories are started in parallel from the initial state with each trajectory assigned a statistical weight. To control the trajectory distribution, configurational space is divided into bins along a 'progress coordinate' toward the target state. Trajectories are evaluated at fixed time intervals $\tau$ for either replication or combination to maintain the same number of trajectories/bin with the goal of generating a sufficiently large ensemble of continuous, successful pathways for computing rate constants. Rigorous management of trajectory weights ensures that no bias is introduced into the dynamics.

WE simulations can be carried out under steady-state or equilibrium conditions. Steady-state WE simulations are carried out by 'recycling' trajectories that reach the target state (i.e., terminating the trajectories and starting new ones from the initial state with the same weights as the terminated trajectories), while equilibrium WE simulations are run with no recycling. Importantly, steady-state trajectories in opposite directions can be combined to generate a set of equilibrium trajectories thereby enabling the estimation of equilibrium observables[16]. In this study, we first carried out equilibrium WE simulations to generate pre-equilibrated initial states and then started steady-state WE simulations from these initial states to generate switching pathways and rate constants.

All WE simulations were carried out using the open-source, highly scalable WESTPA software package (https://westpa.github.io/westpa/)[37] at 20 °C using a $\tau$-value of 100 ps. Each steady-state WE simulation of switching pathways required 3 days using 144 Intel Xeon 2.6 GHz CPU cores in parallel. All analysis was performed using the latter 90% of the simulations with conformations sampled every 50 ps. See Supplementary Methods for full details of the simulation model and WE simulations as well as the calculation of free-energy surfaces, contact scores, simulation efficiency, and number of independent switching events.

**Calculation of rate constants**. Macroscopic rate constants $k_{ij}$ for folding (unfolding) and switching processes involving an initial state $i$ and target state $j$ were computed as follows:[16]

$$k_{ij} = \frac{f_{ij}}{p_i} \qquad (1)$$

where $f_{ij}$ is the flux of probability carried by trajectories originating in state $i$ and arriving in state $j$ and $p_i$ is the fraction of trajectories more recently in $i$ than in $j$. Each state was defined based on equilibrium WE simulations: the boundary of the

state was taken to be the position of the maximum of the probability distribution when viewed as a function of the progress coordinate.

**Equilibrium denaturation experiments**. Protein samples (20 μM) were prepared in 20 mM Tris (pH 7.2), 0.15 M NaCl, 0.1 mM EDTA, 1 mM TCEP, and various concentration of GdnHCl using a Hamilton Microlab 540B Dispenser as described in Stratton et al.[13] Samples were equilibrated at 22 °C overnight and data were collected on an Aviv Model 420 circular dichroism (CD) spectrometer. All CD data was fit to a linear extrapolation equation[38] to yield the folding free-energy ($\Delta G^{fold}$), m-value, and midpoint of denaturation ($C_m$) for each protein of interest.

**Fluorophore labeling and stopped-flow experiments**. WT and mutant AFF switch constructs were cloned, expressed, purified, and labeled with BODIPY-FL as described previously[10]. The $N \rightarrow N'$ and $N' \rightarrow N$ conformational changes shorten and lengthen the distance between the BODIPY groups, causing FRET-induced self-quenching to increase and decrease, respectively. The rate constants $k_{N \rightarrow N'}$ and $k_{N' \rightarrow N}$ were measured by adding $Ca^{2+}$, i.e., mixing 1 mM $CaCl_2$ with the WT E65Q and E65′Q constructs, respectively, and monitoring the change in fluorescence intensity by stopped-flow fluorescence. Proteins were desalted into 2.5 M GdnHCl, 20 mM sodium phosphate (pH 7.0), 0.1 mM EDTA using 10DG columns (Bio-Rad), at which point a 3–5 molar excess of the BODIPY-FL maleimide fluorophore (Molecular Probes) was immediately added. The labeling reaction was allowed to proceed in the dark for 2 h at 25 °C. GdnHCl and unreacted BODIPY-FL were removed by passing the sample through a 10DG column equilibrated in 20 mM Tris (pH 7.5), 0.15 M NaCl, 0.2 mM EDTA, and 0.2 % TWEEN-20. BODIPY-FL concentration was measured by absorbance ($\varepsilon_{503} = 80,000\ M^{-1}\ cm^{-1}$).

Stopped-flow fluorescence data were recorded at 20 °C using Bio-Logic SFM-4000 rapid mixing device and Bio-Logic MOS-200 spectrometer. Excitation was at 487 nm and a 497 nm high-pass filter was used on detection. The fluorescence change was monitored after mixing one volume of BODIPY-FL labeled protein (10 μM) with one volume of 2.1 mM $CaCl_2$ in 50 mM Tris (pH 7.5), 0.1 % TWEEN-20. Data were analyzed using the Bio-Kine software package (Bio-Logic).

Full details of gene construction, protein expression, and protein purification are provided in Supplementary Methods. Primer sequences are provided in Supplementary Table 3.

**Code availability**. The WESTPA software package used in this study is available at https://westpa.github.io/westpa/ for free under the GNU General Public License.

**Data availability**. The data that support the findings of this study are available from the corresponding authors upon reasonable request.

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

## Acknowledgements

This work was supported by NIH 1R01GM115805-01 to L.T.C., NIH 1R01GM115762 to S.N.L., and a University of Pittsburgh Honors College Brackenridge Undergraduate Research Fellowship to A.J.D. Computational resources were provided by NSF CNS-1229064 and the University of Pittsburgh's Center for Research Computing.

## Author contributions

A.J.D., J.H., S.N.L., and L.T.C. designed research, analyzed data, and wrote the manuscript. A.J.D. performed simulations. J.H. and S.N.L. performed experiments.

## Additional information

**Competing interests:** The authors declare no competing financial interests.

