## [Peer Review File · Nature Communications]

Reviewers' comments:

Reviewer #1 (Remarks to the Author):

In this study, the authors present a new simulation-based computational approach to optimize the response time of a protein-based biosensor, the calbindin-AFF switch. They test their predictions experimentally, and find that the mutations identified in their simulations do indeed improve the response time of their sensor, bringing it into the same timescales as fast fluctuations in cytosolic [Ca²⁺]. This reviewer finds the initiative towards rational design exciting and commendable, as biosensor optimization is still often performed by trial-and-error, and the approach the authors present is of general value to the field.

The computational part of the manuscript appears to contain an adequate amount of detail for reproducibility, and most of the main text is accessible to a non-expert audience without additional reading. As an experimentalist, I will, however, focus my evaluation on the experimental work, and on the comparison between experiment and simulation.

The authors' conclusion that their approach involves "preferential stabilization of non-native transient states relative to the ground states...to reduce the free energy barrier for the switching process" appears to be supported by both the theoretical and experimental results in e.g. fig. 3, fig. 4, and Supplementary Table 2. The statistics appear typical for the field and the error bars are defined in the figure legends, although the calculation of the S.E.M. for the $\Delta\Delta G$ values in Table 1 and Supplementary Table 1 would benefit from an additional replicate of the WT N-frame analogue and the WT calbindin to yield $n = 3$. There are also a few aspects of the manuscript that would benefit from additional clarification to improve the general appeal of the study and make the manuscript suitable for publication:

1. The main result of the study is that the authors have successfully optimized the response time of their Ca²⁺ frame-shifting sensor using a new computational approach described in the manuscript. Comparing the experimental results in fig. 4C+D to the enhancements computed from simulations in Supplementary fig. 6, it appears to this reviewer that the experimental result is somewhat the opposite of the prediction. Specifically, the stopped-flow data suggest an equal or greater improvement in the switching kinetics for the E65'Q variants compared to the E65Q variants, while these E65'Q variants show equal or lower enhancement than E65Q variants in the prediction in Supplementary fig. 6. If this is correct, and the authors want to claim that the optimization was done rationally, could the authors comment on the possible origins of this apparent discrepancy in the context of the predictive value of their proposed simulation-guided optimization strategy?

2. On p. 4, the authors state that "The structure of CP-calbindin has not been solved, but previous studies have indicated that it is similar to that of WT calbindin". At a glance, the reference given here (18. [Szebenyi, D. M. E. & Moffat, K. The refined structure of vitamin D-dependent calcium binding protein from bovine intestine. *J. Biol. Chem.* 261, 8761-8777 (1986).]) does not appear to discuss the effects of circular permutation on the structure. The authors could assist the reader by pinpointing what evidence they are referring to, and how this reference supports that their structural model is reasonable. Alternatively, this could perhaps be a typo that should refer to reference 16 that shows e.g. helical CD spectra and maintained equilibrium m -values for the permutant?

Further, the statement that "The only prerequisite is that structures are available for the individual switch components, in this case the WT and CP forms of the protein on which an AFF switch is based." is somewhat confusing, as the authors proceed to explain that only the WT structure is known for their proof-of-principle protein, and that the CP structure is modeled based on the WT structure. Perhaps "...a structure is available for at least one of the switch components...", is a more fitting criterion?

3. As evident from the references, the model protein has been extensively characterized in the past. Generally, however, this reviewer believes the manuscript would appeal to a broader audience and provide a more fluent reading experience if it included more of the experimental results from the present study as SI figures in addition to the tables. Based on the current manuscript it is often not possible, even for an expert reader, to directly evaluate the quality of the experimental data, as most of this data only is presented through tables. Also, it is not possible to assess the origin of the differences in the experimental free energies, as no m -values or transition midpoints are reported. Including such data and figures would assist the reader in identifying e.g. the origin of the slight discrepancy in thermodynamic stability of the WT in the present manuscript compared to reference 9 (Supplementary Table 2), and make for a more compelling argument through the increased transparency that comes with the display of data.

4. The experimental data supporting the statement regarding reversible two-state folding on p. 8 is not presented, only mentioned. To fully support this statement, the denaturation curves should be included e.g. in the SI, together with a brief clarification of how reversibility was assessed.

Reviewer #2 (Remarks to the Author):

The authors use computational design and biophysical experiments to improve the response time of switches designed with the alternate frame folding.

I have a few suggestions for improving the manuscript.

1. I have an issue with the authors' posing of the problem. They state that the property of the switch that has proven to be the most challenging to improve is response time. However, many (most?) switches don't have this problem or at least do not have this problem for the application they are used for (i.e. the response time is fast compared to the changes in concentration of the analyte). I think the authors should point out that the issue the authors are addressing is an issue that is only a problem for a subset of switches. They might also point out that switches designed to function by the alternate frame folding mechanism or other mutually exclusive folding-unfolding mechanisms are more likely to have response time issues, whereas switches using other mechanisms with subtler conformational shifts might be expected to have shorter response times. To be fair, that's just my intuition. I'd be curious if the authors agree with it. Many other switches haven't measured response times because they are not an issue, so it is hard to back this up with hard data.

2. Beginning of third paragraph of introduction. I assume AFF stands for alternate frame folding. It would be helpful to the reader to define the AFF acronym here. Also, reading this paragraph it will be unclear to the reader whether the calbindin-AFF switch was previously developed or is developed for the first time in this manuscript. It sounds like it was previously developed, but there is no citation.

3. Figure 1. Based on the legend, I think b and c should be switched as the panel labels.

4. The author say in the conclusion that their design strategy is a general one. However, I am not sure it could be applied to switches that don't depend on the relative stabilities of two domains. The authors might more explicitly address the following question: Could their methods be applied to switches that function by other mechanisms or is it limited to those that use the alternate frame folding mechanism or other mutually exclusive folding-unfolding mechanisms?

Reviewer #3 (Remarks to the Author):

In this article, the authors have tried to control the response time of protein conformational switch by integrating computational predictions and experimental verifications. In particular, they dealt with calbindin-AFF as a target system, which has typical Ca^{2+} binding modules. They achieved 32-fold switching ratio compared with the wild-type. Thus, the trial may be interesting and useful. However, they only tried several mutations that are not always derived only from the molecular simulation but can be easily obtained by Ala-scan or site-directed mutation on the interface of EF-domains. In particular, they did not truly optimize the response time by using a sort of optimization procedure and/or feedback from experiments. This means that the title of the present article is a bit exaggerated. Therefore, the reviewer has negative toward the publication of the article in Nature Communications. Individual comments are listed below.

(1) In figure 1 b, this protocol is nothing new. How do the authors optimize the function with this protocol? c is also too abstract. Is it always true that free energy levels of N and N' are the same?

(2) Figures 2 d and e are too messy. How the authors specified the position of fluorophores (simulations, experiments, or else)?

(3) Judging from Figure 4-a, there exist other amino acid residues that have high C_N - C_N - N' TPE values. However, the authors only try to mutate Phe replaced by Ala. It is quite an arbitrary choice made by the authors "EMPIRICALLY".

(4) The reviewer does not believe that the residue level Brownian dynamics with weighted ensemble has enough accuracy, i.e. $\Delta\Delta G$ of two digits after the decimal point.

(5) In some averaging procedures, the authors use a different number of samples like as in Table 1 and Figure 6. Please use the same sample numbers.

The reviewers raise a number of essential points and we are indebted to them for their input. We have made extensive modifications to our manuscript (NCOMMS-17-20225A), including results from (i) new simulations and experiments in which we identify and test a negative control mutant (L31A) that is predicted to destabilize the ground and transition states equally (and thus have no effect on switching rates), and (ii) ¹⁵N-HSQC NMR experiments to characterize the structures of both the wild-type and circular permutant forms of the calbindin protein. We feel that these new data, together with the specific responses to each reviewer's comments (below) significantly strengthen the conclusions of our paper and increase its impact.

Reviewer 1

1. Comparing the experimental results in Fig. 4C+D to the enhancements computed from simulations in Supplementary Fig. 6, it appears to this reviewer that the experimental result is somewhat the opposite of the prediction. Specifically, the stopped-flow data suggest an equal or greater improvement in the switching kinetics for the E65'Q variants compared to the E65Q variants, while these E65'Q variants show equal or lower enhancement than E65Q variants in the prediction in Supplementary Fig. 6. If this is correct, and the authors want to claim that the optimization was done rationally, could the authors comment on the possible origins of this apparent discrepancy in the context of the predictive value of their proposed simulation-guided optimization strategy?

We have commented on the possible origins of this apparent discrepancy in the following paragraph, which has been added to the end of the "Tests of computational predictions" section on pg. 11 of Results and Discussion:

"While the main goal of this study was to assess the ability of contact scores derived from simulations of the WT switch constructs to predict positions where mutation improves the response time of the switch, we also assessed how well our computational screen modeled the effect of mutations. The fold-increase in $k_{N \rightarrow N'}$ calculated from simulations of E65Q variants is similar to experiment (Supplementary Fig. 9 and Fig. 4c). In contrast, while simulations of the E65'Q variants correctly predict that the mutations improve the response time of the switch, the simulations do not quantitatively predict the fold-change in $k_{N' \rightarrow N}$ (Supplementary Fig. 9 and Fig. 4d). This discrepancy may be due to the fact that the $N' \rightarrow N$ switching process involves a stepwise mechanism including the formation of an intermediate while the $N \rightarrow N'$ switching process is concerted. To more accurately model the intermediate, a more detailed simulation model with the inclusion of attractive nonnative interactions may be required. Nonetheless, the successful prediction via contact scores of sites where mutation improves the response time demonstrates that the simulation model used in this study is useful for protein engineering purposes, even in a case where nonnative contacts play a substantial role."

2. On p. 4, the authors state that "The structure of CP-calbindin has not been solved, but previous studies have indicated that it is similar to that of WT calbindin". At a glance, the reference given here (18. [Szebenyi, D. M. E. & Moffat, K. The refined structure of vitamin D-dependent calcium binding protein from bovine intestine. *J. Biol. Chem.* 261, 8761-8777 (1986).]) does not appear to discuss the effects of circular permutation on the structure. The authors could assist the reader by pinpointing what evidence they are referring to, and how this reference supports that their structural model is reasonable. Alternatively, this could perhaps be a typo that should refer to reference 16 that shows e.g. helical CD spectra and maintained equilibrium m-values for the permutant?

As R1 pointed out, the original reference was incorrect. We now cite the relevant references that suggested the structures of WT and CP calbindin are similar. To more directly explore that idea we performed ¹⁵N-HSQC NMR experiments of the wild-type and circular permutant forms of calbindin. The results are included in the first paragraph of Results and Discussion:

“...However, circular permutation generally preserves the overall structure of a protein except for minor changes around the sites of permutation and linker addition, and previous NMR²⁷ and circular dichroism¹³ results suggested that the structures of WT and CP calbindin are similar. As a further test, we compared their ¹⁵N-heteronuclear single-quantum correlation NMR spectra (Supplementary Fig. 1a). The majority of cross peaks align with the exception of amino acids 40-45 and 70-75. Mapping these residues onto the structure of WT calbindin confirms that structural differences are limited to the permutation site and the C-terminus of the protein to which the linker is attached (Supplementary Fig. 1b). We therefore constructed a model of CP calbindin *in silico* based on the WT calbindin structure, building in the six-residue loop that links the original N- and C-termini (Supplementary Methods).”

We emphasize that these structural perturbations are precisely what one expects (and generally observes) when comparing structures of WT and circularly permuted proteins.

3. Further, the statement that “The only prerequisite is that structures are available for the individual switch components, in this case the WT and CP forms of the protein on which an AFF switch is based.” is somewhat confusing, as the authors proceed to explain that only the WT structure is known for their proof-of-principle protein, and that the CP structure is modeled based on the WT structure. Perhaps “...a structure is available for at least one of the switch components...”, is a more fitting criterion?

We thank the reviewer for this suggestion and have clarified the second sentence of the *Results and Discussion* to read:

“...The only prerequisite is that structures of the switch components are available, *e.g.*, from x-ray crystallography or homology modeling...”

We have also revised the last paragraph of *Conclusions* to read:

“Our computational design strategy is a general one that can be applied to any protein conformational switch of a similar size (*e.g.*, less than a few hundred amino acids) provided that the switching process occurs on the time scale of <100 s and structures of the switch components are available from either experiment or homology modeling. ...”

4. Generally, however, this reviewer believes the manuscript would appeal to a broader audience and provide a more fluent reading experience if it included more of the experimental results from the present study as SI figures in addition to the tables. Based on the current manuscript it is often not possible, even for an expert reader, to directly evaluate the quality of the experimental data, as most of this data only is presented through tables. Also, it is not possible to assess the origin of the differences in the experimental free energies, as no *m*-values or transition midpoints are reported. Including such data and figures would assist the reader in identifying *e.g.* the origin of the slight discrepancy in thermodynamic stability of the WT in the present manuscript compared to reference 9 (Supplementary Table 2), and make for a more compelling argument through the increased transparency that comes with the display of data.

We have included the following new thermodynamic data: (i) performed additional trials of guanidine denaturation experiments for variants in which $n < 3$ ($n \geq 5$ in all cases now); (ii) included *m*-values and transition midpoints (with statistics) as instructed by R1; (iii) performed reversibility tests of both WT and CP calbindin as directed by R1 (see below); (iv) created and purified the L31A mutant of WT and CP calbindin and characterized them by thermodynamic and kinetic experiments.

The experimental data supporting the statement regarding reversible two-state folding on p. 8 is not presented, only mentioned. To fully support this statement, the denaturation curves should be included *e.g.* in the SI, together with a brief clarification of how reversibility was assessed.

We now include forward and reverse denaturation curves of the N-frame and N'-frame analogues as Supplementary Fig. 7. The caption to this figure reads:

“Unfolding (triangles) and refolding (circles) experiments were performed by diluting either native protein (in buffer) or unfolded protein (equilibrated for 3 h in 6.3 M GdnHCl), respectively, into the indicated final concentrations of GdnHCl. The unfolding and refolding curves are coincident, indicating that folding/unfolding is reversible and at equilibrium for both analogues. See main text Methods for experimental details. Solid lines are best fits to the linear extrapolation equation.”

Reviewer 2

1. I have an issue with the authors' posing of the problem. They state that the property of the switch that has proven to be the most challenging to improve is response time. However, many (most?) switches don't have this problem or at least do not have this problem for the application they are used for (i.e. the response time is fast compared to the changes in concentration of the analyte). I think the authors should point out that the issue the authors are addressing is an issue that is only a problem for a subset of switches. They might also point out that switches designed to function by the alternate frame folding mechanism or other mutually exclusive folding-unfolding mechanisms are more likely to have response time issues, whereas switches using other mechanisms with subtler conformational shifts might be expected to have shorter response times. To be fair, that's just my intuition.

R2 is absolutely correct in that the AFF mechanism is more prone to kinetic traps than conformational switches that employ more subtle structural changes, and we now state this. However, it is also relevant to note that faster switching kinetics is almost never a bad thing, and a great deal of effort has gone into speeding up the rates of some popular genetically-encoded switches. To make this point, we cite examples of other protein-based calcium sensors (e.g. GCaMP), which are still too slow to monitor rapid calcium fluctuations (see excerpt below). We therefore argue that a rational method for increasing response time, such as ours, will be of general utility in the design of conformational switches.

“Response time determines how quickly a biosensor can detect the analyte and whether it can track changes in analyte concentration in real time, and the temporal precision by which functional switches can control cellular pathways. For these applications, one typically wants the turn-on and turn-off rates to be as fast as possible. Usages that emphasize maximum signal change or sustained response instead strive for a slow turn-off rate in order to accumulate signal. It is therefore often desirable to tune the kinetics of protein conformational changes—which occur naturally over a wide range of time scales—to optimize a conformational switch for a given application. For example, calcium concentrations can fluctuate as fast as 10 ms in cells, and the slow response time of existing protein-based calcium sensors continues to hamper studies of rapid calcium signaling processes *in vivo*^{8,9}. It is especially important to be able to accelerate the kinetics of the class of switches described here, because switching rates in this case are limited by protein unfolding events,¹⁰ and these can be slow.”

2. Beginning of third paragraph of introduction. I assume AFF stands for alternate frame folding. It would be helpful to the reader to define the AFF acronym here. Also, reading this paragraph it will be unclear to the reader whether the calbindin-AFF switch was previously developed or is developed for the first time in this manuscript. It sounds like it was previously developed, but there is no citation.

We have revised the beginning of the fourth paragraph of the Introduction (previously the third) to read as follows:

“Here, we have rationally improved the response time of a previously developed, protein-based Ca^{2+} sensor,¹³ using a general computational design strategy in synergistic combination with biophysical experiments....”

We define the AFF acronym in the fifth paragraph:

“The protein-based Ca^{2+} -sensor that we examine, calbindin-AFF, was engineered using the alternate frame folding (AFF) scheme.¹³ ...”

3. Figure 1. Based on the legend, I think b and c should be switched as the panel labels.

We have corrected the figure legend to read:

“**b,c**, The strategy for improving the response time of the switch involves **(b)** a close interplay of experiment and computation to **(c)** obtain preferential stabilization of non-native transient states relative to the ground states (N and N') (dashed lines), reducing the free energy barrier for the switching process..”

4. The authors say in the conclusion that their design strategy is a general one. However, I am not sure it could be applied to switches that don't depend on the relative stabilities of two domains. The authors might more explicitly address the following question: Could their methods be applied to switches that function by other mechanisms or is it limited to those that use the alternate frame folding mechanism or other mutually exclusive folding-unfolding mechanisms?

Thank you for the suggestion. We have added the following sentence to the final paragraph of the Conclusion to address the reviewer's question:

“...Furthermore, since all protein conformational switches function based on the relative stabilities of alternate conformations, our strategy is applicable to all such switches, including ones that function by other mechanisms that do not involve as large conformational transitions as the mutually exclusive folding of protein domains, provided that the expected relative stabilities are reproduced...”

Reviewer 3

General comment 1: They did not truly optimize the response time by using a sort of optimization procedure and/or feedback from experiments. This means that the title of the present article is a bit exaggerated.

We have revised the title of the manuscript, which now reads:

“Large enhancement of response times of a protein conformational switch by computational design”

General comment 2: They only tried several mutations that are not always derived only from the molecular simulation but can be easily obtained by Ala-scan or site-directed mutation on the interface of EF-domains.

The value of our strategy is that it is a rational approach that eliminates the need for “trial-and-error” approaches such as alanine scanning. To further demonstrate the power of our computational design strategy, we devised a particularly informative negative control in which we identify a residue (L31) that forms many contacts in the ground states (just like the Phe to Ala rate-accelerating mutants), but retains most of these contacts in the transition path. The L31A mutation

is therefore predicted to destabilize the ground states, but have little effect on the switching rates. The rationale is summarized in the last paragraph of the “Computational predictions to improve kinetics” section on pg. 10:

“An important negative control is to select a mutation that is predicted to destabilize both the ground and transition states to approximately equal extents, and to verify that the switching rates do not change. Among the residues buried at the interface of the EF-1 and EF-2/EF-2’ hands, including the promising four Phe residues, L31 in EF-1 exhibited the lowest change in contact scores for both the N→N’ and N’→N switching processes and was therefore selected as the residue to mutate as a negative control. Given that EF-1 is shared between the N and N’ folds, only a single mutation needs to be introduced into calbindin-AFF, and its effect on switching rate is expected to be symmetric in each direction. As expected, the L31A mutation destabilizes the N- and N’-frame analogues to a comparable extent to the promising mutations, *e.g.* F50A (Table 1).”

We have presented the additional results under the “Tests of computational predictions” section on pg. 11:

“In contrast to the above results, the L31A negative control construct increased the forward and reverse switching rates of WT calbindin-AFF by only 1.03- and 1.57-fold, respectively (Fig. 4c-d). Like the Phe residues, L31 is buried in a tightly-packed environment such that its mutation to Ala significantly destabilizes calbindin. That the L31A mutation has little effect on switching rates highlights the power of the simulations to identify rate-accelerating mutations that would not be obvious from inspecting ground-state structures.”

1. In figure 1 b, this protocol is nothing new. How do the authors optimize the function with this protocol? c is also too abstract. Is it always true that free energy levels of N and N’ are the same?

As mentioned above, we have revised the title of the manuscript such that “optimization” has been replaced with “large enhancement.” While the simulation protocol itself is not new, our study reports the first (to our knowledge) successful rational enhancement of switch response times using this protocol in close interplay with experiment. To emphasize this point, we have revised the last sentence of the third paragraph on pg. 2 to read:

“To our knowledge, the successful rational enhancement of protein switch response times by a computational design strategy has not been reported until now.”

We have also revised Fig. 1b to indicate the use of “WE simulations” since the WE strategy is critical for enabling the simulation of the long-timescale switching events even with the use of residue-level protein models. To clarify Fig. 1c, we have revised the schematic free energy landscapes to be consistent with the E65’Q switch construct in the presence of Ca²⁺ and added the following to the figure caption:

“Panel c depicts a schematic free energy landscape for the E65’Q construct in the presence of Ca²⁺; the strategy is applied identically to the E65Q construct, which differs in that N’ is more stable than N in the presence of Ca²⁺.”

2. Figures 2 d and e are too messy. How the authors specified the position of fluorophores (simulations, experiments, or else)?

We have revised Fig. 2d-e such that only one representative conformation is highlighted and the other conformations of the ensemble are included with decreased opacity to depict the disorder of

the orphan EF-hand while more clearly indicating the positions of the fluorophores. We have also revised the figure caption to read:

“... (see Fig. 1a for color scheme; yellow spheres indicate positions at which fluorophores were inserted in the stopped-flow fluorescence experiments; fluorophores were not included in the simulations).”

3. Judging from Figure 4-a, there exist other amino acid residues that have high C_N-C_N-N[′]TPE values. However, the authors only try to mutate Phe replaced by Ala. It is quite an arbitrary choice made by the authors “EMPIRICALLY”.

The mutation sites were chosen according to the criteria detailed in the manuscript, including no disruption of Ca²⁺ binding. As we state in the manuscript:

“While residues V61 and E60 in the EF-2 hand also exhibited large differences in contact scores between the ground state N and the N→N[′] TPE, these residues are in the vicinity of the Ca²⁺-binding residues and were therefore not mutated.”

Please also see our response to General Comment 2 by Reviewer 3.

4. The reviewer does not believe that the residue level Brownian dynamics with weighted ensemble has enough accuracy, i.e. Delta-Delta G of two digits after the decimal point.

We have revised Table 1 and Supplementary Tables 1 and 2 to report the free energies and rate constants to one digit after the decimal point.

5. In some averaging procedures, the authors use a different number of samples like as in Table 1 and Figure 6. Please use the same sample numbers.

We have performed additional equilibrium denaturation experiments such that the number of replicates is consistent among the constructs shown in Table 1. The intention of the simulations in Supplementary Fig. 9 (which was previously Supplementary Fig. 6; the main text has no Fig. 6) was to provide quick screens of promising mutants. Therefore, the number of replicates for these quick screens of mutant constructs ($n = 3$) is fewer than those carried out for the wild-type constructs ($n = 10$). We have added the following sentence to the captions of Supplementary Fig. 8 and Supplementary Fig. 9:

“To enable quick screening of promising mutations, fewer independent WE simulations ($n = 3$) were performed for mutant constructs than for WT constructs ($n = 10$).”

REVIEWERS' COMMENTS:

Reviewer #1 (Remarks to the Author):

I believe that the authors have addressed the comments brought forward by the referees in a satisfactory manner and that they have significantly improved the readability of the manuscript. In particular, the inclusion of the experimental data is a welcome addition that greatly improves the readers' ability to evaluate the results.

In line with the comments of more than one reviewer, the revised manuscript now clearly describes the authors' computational design strategy as a qualitative tool to predict mutations that increase switching rates. The inclusion of the L31A negative control corroborates this conclusion. The discussion of why the computational design predicts the greatest effect on the E65Q variants, while experiments show that the mutations in fact have a greater effect in the E65'Q variants is also improved. While the authors still are not able to demonstrate the cause of this quantitative discrepancy, their proposed explanation appears plausible in light of the results presented in the manuscript.

This leaves us with a qualitative tool, that potentially can be a time-saver compared to existing experimental strategies such as Ala scans. I believe the authors' claims are reasonable, and that the work is suitable for publication.